

# Modeling unsteady loads on wind-turbine blade sections from periodic structural oscillations and impinging gusts

Nathaniel J. Wei[1,2*] and Omkar B. Shende[3*]

[1]Graduate Aerospace Laboratories, California Institute of Technology, Pasadena, CA 91125, USA
[2]Department of Mechanical Engineering and Applied Mechanics, University of Pennsylvania, Philadelphia, PA 19104, USA
[3]Department of Mechanical Engineering, Stanford University, Stanford, CA 94305, USA
[*]These authors contributed equally to this work.
Correspondence: Omkar B. Shende (oshende@stanford.edu)

**Abstract.** Many traditional methods for wind turbine design and analysis assume quasi-steady aerodynamics, but atmospheric flows are inherently unsteady and modern turbine blades are susceptible to aeroelastic deformations. This study therefore evaluates the effectiveness of simple analytical models for capturing the effects of such unsteady conditions on wind-turbine blades. We consider a pitching and plunging airfoil in a periodic transverse gust as an idealization of unsteady loading scenarios on a blade section. A potential-flow model derived from a linear combination of canonical problems is proposed to predict the unsteady lift on a two-dimensional airfoil in the small-perturbation limit. We then perform high-fidelity two-dimensional numerical simulations of a NACA-0012 airfoil over a range of periodic pitch, plunge, and gust disturbances, and quantify the amplitude and phase of the unsteady lift response. Good agreement with the model predictions is found for low to moderate forcing amplitudes and frequencies, while deviations are observed when the angle-of-attack amplitudes approach the static flow-separation limit of the airfoil. Potential explanations are given for the cases in which the ideal-flow theory proves insufficient. This theoretical framework and numerical evaluation motivate the inclusion of unsteady flow models in design and simulation tools in order to increase the robustness and operational lifespans of wind turbine blades in real flow conditions.

## 1 Introduction

Wind turbines frequently encounter unsteady flow disturbances caused by axial gusts, wind shear and veer across the rotor plane, turbine yaw misalignment, atmospheric turbulence, and myriad other factors that can induce potentially harmful unsteady loads on the turbine blades. As wind turbines continue to increase in size, their blades are becoming longer, lighter, and more flexible, and are thus increasingly susceptible to unsteady flows that drive aeroelastic interactions and structural oscillations. These dynamics can result in increased fatigue loading, decreased operational lifespans, and in extreme cases, catastrophic failure. Parameterizations of the unsteady aerodynamics of gust encounters and turbine blade oscillations are therefore of critical importance to wind-turbine design.

While atmospheric flows are inherently unsteady, many modern tools for wind-turbine design generally treat blade-section aerodynamics in a quasi-steady manner. Traditional blade-element momentum (BEM) codes rely on equilibrium relations between inflow conditions and resultant forces and moments. Actuator-line methods, used in high-fidelity simulations of wind





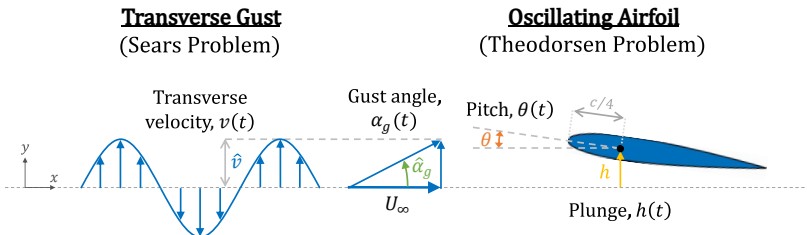

**Figure 1.** Schematic of the superposed gust-oscillation problem: a transverse-velocity perturbation impinging on an airfoil oscillating in pitch and plunge about the quarter-chord point. For a wind-turbine blade section, the convective gust represents a temporal or spatial inflow disturbance, while the airfoil oscillations represent blade deformations due to aeroelastic effects. The transverse gust convects at the free-stream velocity ($U_\infty$) and oscillates with frequency $f_g$, while the airfoil oscillates with frequency $f_p$. Circumflexes denote amplitudes.

turbines and their wake dynamics, typically rely on this approach as well (e.g. Churchfield et al., 2017). Free-vortex wake
(FVW) solvers can capture unsteady wake dynamics at the cost of increased computational complexity, but often rely on quasi-steady blade-element descriptions of the forces on each blade section to match the associated bound circulations (e.g. Marten et al., 2015). Unsteady inflow conditions are often represented using dynamic inflow models (e.g. Snel and Schepers, 1995; Henriksen et al., 2013), which generally involve empirically tuned relations for the time response of the blade sections (Yu et al., 2019). While analytical models for the time response of the full turbine rotor have recently been derived and experimentally
validated (e.g. Mancini et al., 2020; Wei and Dabiri, 2023), these have not yet been connected to unsteady aerodynamics at the blade-section level. In the case of blade sections undergoing dynamic stall in large-amplitude disturbances, the semi-empirical model of Leishman and Beddoes (1989) may be employed at a blade-section level. These dynamic models are widely used in turbine design tools (e.g. AeroDyn and QBlade), but for general unsteady flow-structure interactions, a fully analytical framework may provide richer physical insights into the blade-section dynamics and loads.

Fortunately, gust encounters and blade-oscillation problems have long been treated in classical aerodynamics theory using potential-flow arguments within a linear framework (cf. Jones, 2020). The canonical model for the force on an airfoil in a small-amplitude transverse gust was first derived by Sears as a transfer function (Sears, 1938; von Kármán and Sears, 1938). This result, known as the Sears function, represents the ratio of the unsteady lift amplitude to the equivalent quasi-steady lift amplitude. It predicts a decreasing lift response with increasing reduced frequency, defined as $k_g = \pi \frac{f_g c}{U_\infty}$, where $f_g$ is the
gust frequency (in Hz), $c$ is the airfoil chord, and $U_\infty$ is the free-stream velocity. This scenario is relevant to wind turbines when the blade encounters fluctuations in the incoming flow. In a similar manner, the force on an airfoil subject to pitch and plunge oscillations at a reduced frequency $k_p$, was characterized by Theodorsen to yield what is known as the Theodorsen function (Theodorsen, 1934). For a wind turbine, these oscillations may represent blade-section motions that occur as the turbine blade flexes along its span due to aeroelastic effects. Both Sears and Theodorsen models have been used extensively in
the aerodynamics community (Leishman, 2006), alongside similar approaches by Wagner (1925), Küssner (1936), Greenberg (1947), and others. Extensions of these problems to include higher-order effects such as nonzero mean angle of attack and





airfoil camber (Goldstein and Atassi, 1976; Atassi, 1984), three-dimensional effects (Massaro and Graham, 2015), and the effects of airfoil thickness (Lysak et al., 2013) have also been derived. Furthermore, both the Sears and Theodorsen models have been extensively validated in experiments (e.g. Baik and Bernal, 2012; Wei et al., 2019; Li et al., 2022). Deviations

from their predictions are seen at high forcing frequencies (Taha and Rezaei, 2019; Catlett et al., 2020) and large airfoil-oscillation amplitudes (Baik and Bernal, 2012). Still, even in conditions that violate the small-amplitude assumptions of the theory, qualitative predictions of unsteady lift can still be obtained (Baik et al., 2012; Brunton and Rowley, 2013). Hence, the framework shared by the Sears and Theodorsen functions may yield analytic insights into the unsteady aerodynamics of two-dimensional (2D) turbine-blade sections in gusts and oscillatory motions.

The idealized disturbance kinematics of the Sears and Theodorsen problems, however, cannot represent the full range of gust-disturbance and airfoil-oscillation modes that may be experienced by a turbine blade section in atmospheric turbulence, particularly in the case of simultaneous disturbances from gusts and forced oscillations. A line of inquiry that captures these additional effects without abandoning the elegance of the linear potential flow approach is to superimpose the two forcings to study the combined scenario of a pitching and plunging airfoil in a periodic transverse gust. This formulation, shown

schematically in Figure 1, represents a more realistic problem than either component in isolation. McGhee et al. (2022) recently adopted a similar approach in OpenFAST simulations of a full wind turbine, analyzing aeroelastic blade deformations in response to sinusoidal temporal and spatial inflow disturbances. The linear functional form of the superposed problem also means that it can be directly generalized to real flow scenarios for load control and disturbance rejection. The problem can additionally serve as an idealization of the aeroelastic blade-section dynamics of a yaw-misaligned turbine (Micallef and Sant,

2016; Schiffmann, 2020).

Perhaps surprisingly, the dynamics of this superposed gust-oscillation problem have only been explored in a few studies (e.g. Lian and Shyy, 2007; Webb et al., 2008; Golubev et al., 2010; Ma et al., 2021). Leung et al. (2018) show some equivalence between controlled airfoil-oscillation kinematics and the lift response from a transverse gust at reduced frequencies below $k = 0.75$, which suggests that these two unsteady forcing mechanisms involve similar underlying physics. To our knowledge,

however, there exists no theoretical work on the combined effect of transverse gusts and airfoil oscillations on unsteady lift, and the complexity and expense of generating controlled transverse gust disturbances in experiments makes this problem particularly difficult to study in a laboratory setting.

Therefore, this study aims to evaluate the ability of classical frameworks to capture realistic unsteady loads on a representative 2D turbine blade section. We adopt two key simplifying assumptions in this work. First, the superposed gust-oscillation

problem serves as an idealization of the dominant forms of unsteady loading on a turbine blade at the 2D section level. Additional effects such as airfoil thickness, camber, nonzero mean angle of attack, and finite span can be accounted for as corrections to the idealized framework by drawing from the unsteady-aerodynamics literature. Second, we constrain our investigations to 2D high-fidelity numerical simulations of the superposed gust-oscillation problem. This matches the 2D assumptions of both the potential-flow model and the blade-element methods that are the intended beneficiaries of this work, allowing the accuracy

of the model to be tested on its own terms and in the context of its ultimate applications. Though this study does not then





implement the unsteady modeling framework in a blade-element code, the evaluation of the model presented here serves as a necessary validation step for future unsteady extensions of blade-section methods for wind-turbine design and analysis.

The work is organized as follows. In Section 2, a theoretical model is derived that results in a linear combination of the Sears and Theodorsen functions. The computational framework is described in Section 3, and results from the simulations are
compared with the model predictions in Section 4. Finally, the implications of this parametric study for aerodynamic control, modeling, and design applications are discussed in Section 5.

## 2 Theoretical Model

To derive a model for the unsteady lift force on an airfoil undergoing periodic pitching and plunging motions in a periodic transverse gust, we follow the general approach of Sears (1938), von Kármán and Sears (1938), and Sears (1941). We assume
the flow is 2D, incompressible, and inviscid, such that potential-flow modeling techniques may be employed, and model the wake of the airfoil as a smooth 2D sheet of shed vorticity.

Both Theodorsen (1934) and Sears (1941) evaluated the unsteady lift as derived by von Kármán and Sears (1938) for a pitching and plunging airfoil. Defining the pitch ($\theta$) and plunge ($h$) kinematics as small-amplitude sinusoidal perturbations referenced to the quarter-chord point (as shown in Figure 1), i.e. $\theta(t) = -i\hat{\theta}e^{if_p t}$ and $h(t) = -i\hat{h}e^{if_p t}$, the lift force can be
written as a function of the pitch and plunge amplitudes ($\hat{\theta}$ and $\hat{h}$), the nondimensional pitch-axis location with respect to the mid-chord point ($a$, where $a = -1$ represents the leading edge), and the reduced frequency of the oscillations ($k_p$):

$$L_p = \frac{1}{2}\pi\rho c U_\infty^2 e^{if_p t}\left[-ik_p\hat{\theta} - 2i\frac{\hat{h}}{c}k_p^2 - a\hat{\theta}k_p^2 + 2C(k_p)\left(-\hat{\theta} + 2i\frac{\hat{h}}{c}k_p + i\left(a - \frac{1}{2}\right)\hat{\theta}k_p\right)\right]. \tag{1}$$

The complex function $C(k)$ is written in terms of modified Bessel functions of the second kind, $K_\nu(x)$, with order $\nu$ as

$$C(k) = \frac{K_1(ik)}{K_0(ik) + K_1(ik)}. \tag{2}$$

While the work of both Sears and Theodorsen include this derivation, we will refer to Equation 1 as the Theodorsen function.

Sears (1941) performed a similar analysis for a stationary airfoil at zero mean angle of attack in a periodic transverse gust with vertical-velocity profile $v_g(t) = \hat{v}_g e^{if_g t}$ as

$$L_g = \pi\rho c U_\infty \hat{v}_g e^{if_g t}\frac{J_0(k_g)K_1(ik_g) + iJ_1(k_g)K_0(ik_g)}{K_1(ik_g) + K_0(ik_g)}, \tag{3}$$

where $J_\nu(x)$ are modified Bessel functions of the first kind with order $\nu$. We will refer to this expression as the Sears function,
and it can also be parameterized by the gust-angle amplitude, $\hat{\alpha}_g = \tan^{-1}(\hat{v}_g/U_\infty)$.

The lift force is traditionally represented in nondimensional form as the sectional lift coefficient, $C_l = L/\frac{1}{2}\rho c U_\infty^2$. Alternatively, the unsteady lift can be normalized by the quasi-steady lift,



$$L_{qs} = \frac{1}{2}\rho c U_\infty^2 \left( 2\pi \tan^{-1}(\hat{v}_g/U_\infty) \right) e^{if_g t} \approx \pi \rho c U_\infty \hat{v}_g e^{if_g t}, \tag{4}$$

which represents the equivalent lift amplitude under the assumption that the lift is solely a function of the instantaneous angle
of attack ($\alpha$) and varies as $C_l = 2\pi\alpha$ in accordance with 2D thin-airfoil theory. This normalization gives a transfer-function
form, $h = L/L_{qs}$, which yields gain and phase predictions for a periodic disturbance.

In brief, the linear nature of this modeling framework implies that the full problem can be modeled by a combination of
the Sears and Theodorsen functions. By adding Equations 1 and 3, we obtain a potential-flow model for the unsteady lift of
an oscillating airfoil in a periodic transverse gust, i.e. $L_{dyn} = L_p + L_g$. Note that this function is formulated for sinusoidal
variations in pitch and plunge and a cosine profile for the gust velocity. Adjustments can be made to represent other phase
relationships between the three perturbations. For both components, any disturbance with nonzero frequency returns a gain
that is less than one, implying that, in the small-amplitude regime, no amplification is expected.

Though the unsteady lift in this combined model involves multiple forcings, the effective (net) angle of attack ($\alpha_{eff}$) for a
set of gust and airfoil-oscillation kinematics following Xu and Lagor (2021) is given by

$$\alpha_{eff} = \sin(\theta) + \frac{1}{2}\left(\frac{1}{2} - a\right)\frac{c\dot{\theta}}{U_\infty} - \frac{\dot{h}}{U_\infty}\cos(\theta) + \frac{v_g}{U_\infty}\cos(\theta), \tag{5}$$

where $\dot{\theta}$ and $\dot{h}$ are the pitch and plunge rates. This provides a quasi-steady normalization based on the steady lift polar.

At high amplitudes and frequencies of forcing, viscous effects are non-negligible, so once the nominally 2D wake assumption
becomes invalid and vortex stretching becomes active in the full three-dimensional problem, the approach used here becomes
less appropriate. As such, it is only strictly applicable prior to any signs of transition and stall on the airfoil. Even at lower
perturbation amplitudes, high reduced frequencies may introduce viscous effects as the planar wake assumed by the potential-
flow model begins to roll up in a nonlinear fashion. Still, for relatively low amplitudes and frequencies, we expect the model
to capture the leading-order physical phenomena that dominate the dynamics of the system. Additionally, as mentioned in the
previous section, several corrections for non-ideal airfoil effects (e.g. thickness, camber, etc.) exist in the literature, and the
linear nature of this modeling framework suggests that these extensions could be integrated into the present model as well.

Neither this model nor the following simulations consider the effects of nonlinear coupling between impinging gusts and
airfoil oscillations, including gust-induced aeroelastic effects such as flutter. Such phenomena are related in principle to the
unsteady dynamics considered in this study, but since we only consider linear combinations of the transfer functions derived
from Equations 1 and 3, and their individual gains are less than one for all nonzero frequencies, the combined model cannot
predict phenomena associated with coupling, resonance, or amplification of disturbances, which we consider to be outside the
scope of this work. We constrain our investigations to two independent sources of unsteady lift (i.e. airfoil oscillation and the
impinging gust) and remain agnostic to the origins of and interactions between these two modes.

These considerations notwithstanding, some noteworthy implications can still be drawn from this model. The superposition
of two harmonic signals at different frequencies creates a long-period oscillation with a beat frequency given by the difference





of the two forcing frequencies. In the present problem, this occurs when $f_g \neq f_p$. In these scenarios, constructive and destructive

interference of the input waveforms periodically modifies the amplitude of the net lift. The exact phase relationship between the gust and airfoil oscillations is therefore critical to predicting the short-period lift amplitude in practical applications, particularly when the forcing frequencies are very close together. When the frequencies are identical, however, the beat period is eliminated, and only a single phase relationship is maintained. Thus, the amplitude will differ from that computed for a neighboring frequency ratio (e.g. $f_g/f_p = 0.999$) over its full beat period, and changing the phase between the forcing inputs will change

the observed lift amplitude. In the limiting case where one forcing frequency is zero, there is no beat oscillation that can modify the amplitude, and the amplitude becomes decoupled from the phase of the input waveform. The ability to capture these nontrivial frequency dependencies is a strength of this modeling framework, and the transfer-function form of the model is particularly well-suited for control applications in unsteady aerodynamics.

## 3 Methods

To investigate the performance of the proposed modeling framework, we employ simulations of the filtered compressible Navier-Stokes equations in the isothermal low-Mach number limit. As we fully resolve the largest spatiotemporal flow scales, this is a higher-fidelity approach than unsteady Reynolds-averaged Navier-Stokes simulations for accurately simulating highly unsteady flows and moving boundaries. The 2D setup matches the framework's theoretical assumptions, which consider an infinite-span airfoil, but may not quantitatively match turbulent statistics obtained from a laboratory experiment due to the

different energetics involved. However, in the limit of small perturbation amplitudes consistent with potential-flow assumptions, it should reasonably capture the underlying physics prior to stall. Lastly, the reduced computational cost of 2D simulations relative to 3D ones allows a much wider parameter space to be investigated, particularly since this study is focused on the integrated quantity of lift and not specific details of the flow kinetics. This choice limits the ability of these results to speak to dynamics outside of the small-amplitude perturbation regime, including flow separation, dynamic stall, and vortex shedding.

We use NaluCFD, a massively parallel solver for the low-Mach Navier-Stokes equations, whose variants have been verified and validated for a wide range of applications, including wind and atmospheric modeling (Domino, 2015; Kaul et al., 2020). Two additional source terms are implemented alongside the standard governing momentum equations: the first governs mesh motions that represent airfoil pitch, and the second adds a body force to the entire domain to represent plunging motions. An adaptive time-stepping method is used to properly resolve unsteadiness, while the use of the low-Mach equation set allows

time steps much larger than the acoustic timescale. Details regarding the construction, verification, and validation of the base solver and the boundary condition sets are provided by Domino (2015). This section will focus on the simulation domain, modifications made to the solver, and the design of the numerical experiments of this study.

### 3.1 Computational Details

The basic structure of the domain used for the present study is shown in Figure 2. The domain is rectangular, with dimensions

of $8c$ in the vertical direction and $6c$ in the horizontal. Since vortical structures shed from the airfoil will generally have





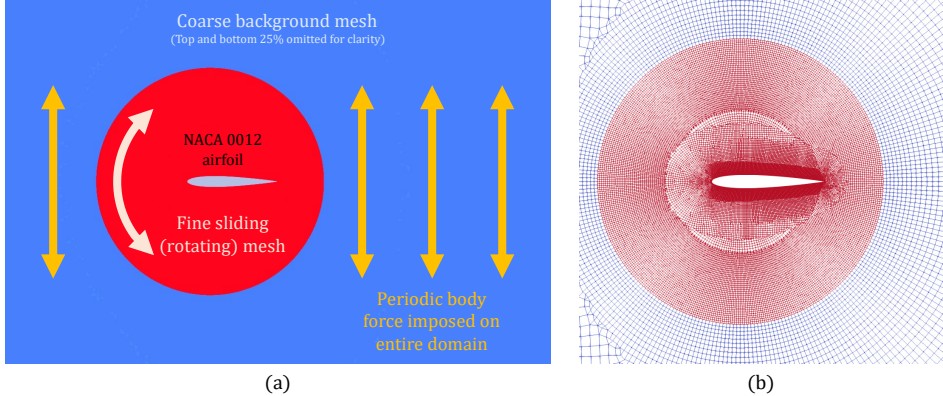

(a)      (b)

**Figure 2.** (a) Diagram of the computational domain, composed of a coarse background mesh (blue) and a refined sliding circular mesh (red) containing the NACA-0012 airfoil (grey). The top and bottom 25% of the domain are omitted for clarity. The circular mesh is rotated to simulate pitching motions, and a body force (orange) is imposed on the entire domain to represent plunging motions. (b) Close-up representation of the grid, showing the coarse background mesh (blue) and the refined circular mesh containing the airfoil (red).

length scales less than $1$ or $2c$, this domain is large enough to mitigate boundary-condition effects on the dynamics we are studying. Periodic boundary conditions are enforced on the top and bottom boundaries of the domain, a prescribed laminar inflow condition is given on the inlet boundary, and an open convective characteristic boundary condition is sustained on the outflow side. The solution was found to be insensitive to the choice of a symmetry boundary condition instead of a periodic condition for the top and bottom boundaries. Inside the domain, a NACA-0012 airfoil is embedded in a circular rotating mesh with a radius of $1.25c$, which allows for the simulation of airfoil pitch motions.

Details of the boundary condition treatments are provided in (Domino, 2015). In general, a second-order central-difference spatial discretization is utilized in the bulk domain and upwinding is used at the non-conformal interface with Gauss-Lobatto quadrature to handle flux translations between the two mesh regions. To implement pitch for the airfoil, a sliding mesh with a non-conformal interface (as implemented in NaluCFD using discontinuous Galerkin methods) is used.

The unstructured grid is meant to resolve the boundary layer expected at the airfoil surface, highlighted in Figure 2. While not significant in this potential flow analysis, remaining unresolved effects in the filtered Navier-Stokes equations are captured by the Wall-Adapting Local Eddy-viscosity (WALE) model (Ducros et al., 1998). This algebraic model, formulated to provide closure terms for large-eddy simulations, captures the asymptotic behavior of small-scale energetics (i.e. the expected eddy viscosity scaling near the wall, cf. Nicoud and Ducros, 1999) for wall-bounded flows. As primary interest is in unsteady phenomena at length scales approaching the airfoil chord length, wall-resolution effects should not significantly affect results.

In lieu of implementing mesh motion via translations of the underlying overset or deforming meshes for the large-scale motions needed for plunging, we instead move the bulk fluid directly. This is equivalent to transforming into a non-inertial frame that is fixed to the airfoil, as done in other studies (Ramos et al., 2019). This project, therefore, concerns results that are given in a non-inertial frame, and so we add a frame-acceleration term to the momentum equation to perform this transform.



This source term is based on a specified plunge-motion profile and is added directly to the right-hand side of the momentum equation. The plunging motion of the airfoil is given as $h(t) = \hat{h}\sin(2\pi f_p t)$, where $f_p$ is the plunging frequency (in Hz). Therefore, the source term is represented simply as the acceleration, $\ddot{h}(t)$.

The primary non-standard boundary condition is for the inflow, where a sinusoidal gust in the vertical velocity can be fed into the domain along with a constant mean velocity in each component. This inflow is not turbulent. As the vertical velocity must be with respect to an inertial reference frame, one must also supply information about the momentum source term to correct the sinusoidal inflow. Since the bulk velocity is integrated from an acceleration source term, while the velocity at the boundary is specified directly, a slight numerical inconsistency between the boundary condition and bulk fluid velocity next to the boundary is unavoidable. This small discrepancy exists in phase and amplitude near the boundary and advects through the domain; however, its magnitude is negligible relative to the unsteady dynamics in the simulation.

## 3.2 Characterization and Analysis of Cases

Around fifty test cases are simulated using the aforementioned computational apparatus at a chord-based Reynolds number of $Re_c = \frac{cU_\infty}{\nu} = 2.22 \times 10^5$, where the chord length $c = 1$ m and the inflow velocity $U_\infty = 4\,\mathrm{ms}^{-1}$. This $Re_c$ is high enough that viscous effects neglected in the analytical framework do not dominate the flow physics, but low enough so that the nonlinear effects of high Reynolds-number turbulence are limited, according to guiding principles of airfoil design (Lissaman, 1983). This also corresponds to $Re_c$ often observed in wind-tunnel tests of similar problems (e.g. Wei et al., 2019). Other non-dimensional parameters of interest to this flow situation are the reduced frequency and the Strouhal number $St_p = \frac{f_p A}{U_\infty}$, where $A$ represents the trailing-edge amplitude of the airfoil (Anderson et al., 1998). As the pitch amplitude is generally small in these simulations, $A \approx \hat{h}$. In this study, we investigate reduced frequencies in the range of $0.196 \leq k_g \leq 3.93$ for the gust disturbance and $0.785 \leq k_p \leq 3.93$ for the airfoil oscillations. These reduced frequencies are relatively high compared to typical turbine-blade fundamental frequencies and apparent gust frequencies due to nonuniform inflow conditions (e.g. wind shear and veer), which generally yield $k \lesssim 0.2$ (cf. Hansen et al., 2006; Sebastian and Lackner, 2013). Still, wind turbines operating in off-design or highly turbulent conditions may experience disturbances with reduced frequencies approaching unity (Boye and Xie, 2022). The range of $St_p$ investigated, by contrast, is relatively low ($0.008 \leq St_p \leq 0.020$), in keeping with the assumption that the amplitudes of oscillation for turbine blades under typical aeroelastic forcings will not be extreme. Thus, in the limit of small-amplitude disturbances where large vortical structures are not shed from the airfoil and the wake width remains thin relative to the blade section chord, we do not expect $St_p$ to play a significant role in the dynamics (cf. Anderson et al., 1998).

Several combinations of parameters are tested to explore the parameter space along different axes. First, the frequency ratio between the gust and airfoil-oscillation frequencies is varied between $f_g/f_p = 0.25$ and 5, with the gust-angle, pitch, and plunge amplitudes held constant at $\hat{\alpha}_g = \hat{\theta} = 3°$ and $\hat{h}/c = 0.03$, respectively. For convenience, we will refer to these data as **FR** cases. A frequency ratio of $f_g/f_p = 0$ corresponds to a purely oscillating airfoil, whereas a frequency ratio of $f_g/f_p \to \infty$ corresponds to a stationary airfoil in a gust. We are particularly interested in the interactions between these two effects, parameterized by frequency ratios near unity. These are repeated with a $180°$ phase shift in the gust disturbance, which we call **FRn** cases. Additionally, several tests are conducted at a fixed frequency ratio of $f_g/f_p = 1$: the gust amplitude is





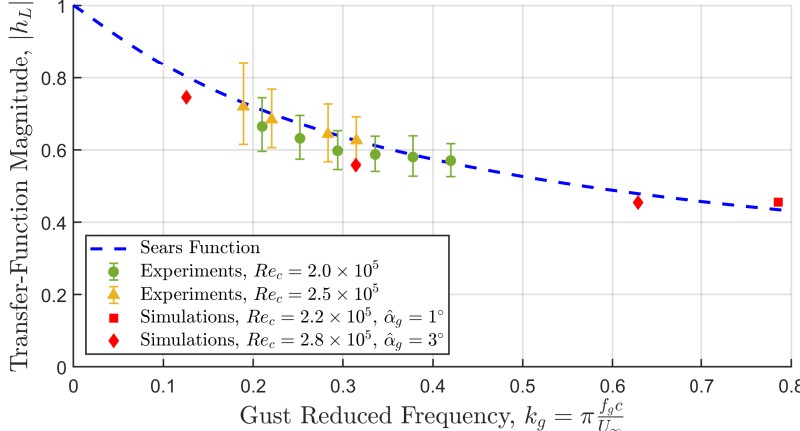

**Figure 3.** Unsteady lift-response data of a stationary airfoil in a periodic transverse gust from numerical simulations at $Re_c = 2.22 \times 10^5$ and $Re_c = 2.78 \times 10^5$, plotted against gust reduced frequency $k_g$ and compared with the Sears function. Experimental results by Wei et al. (2019), which tested a NACA-0006 profile at $Re_c = 2.0 \times 10^5$ and $2.5 \times 10^5$, are also provided for context.

varied between $0° \leq \hat{\alpha}_g \leq 8°$ (with the pitch and plunge amplitudes held constant at $\hat{\theta} = 1°$ and $\hat{h}/c = 0.01$); the pitch and plunge amplitudes are increased proportionally from zero to $\hat{\theta} = 8°$ and $\hat{h}/c = 0.08$ (with the gust amplitude held constant at $\hat{\alpha}_g = 1°$); and finally, all three amplitudes are increased proportionally from zero to $\hat{\alpha}_g = \hat{\theta} = 8°$ and $\hat{h}/c = 0.08$. These are referred to as **G**, **PP**, and **PPG** cases, respectively. In the small-amplitude limit where the inviscid theory remains valid, increasing amplitudes in this manner should not affect the accuracy of the theoretical predictions.

In every case, the simulation is run for 32 cycles of the lowest-frequency forcing. The lift force is computed by integrating pressure over the airfoil surface and the first unsteady-forcing period is rejected from these data to remove start-up effects. The remaining cycles are phase-averaged over the least common multiple of the two forcing periods and the equivalent theoretical lift profile is obtained by computing values for the theoretical lift curve over this interval. For cases where the frequencies are not identical, this leads to a long-period response driven by the beat frequency of the linearly superposed forcings. The

phase-averaged simulation results are displayed with uncertainty intervals corresponding to one standard deviation about the mean; these represent the cycle-to-cycle variability of the lift force across all simulated cycles.

The RMS value of the phase-averaged lift profile provides the amplitude of the unsteady lift response, and the phase is obtained from the argument of the frequency corresponding to the least-common-multiple forcing period, given by a fast Fourier transform. The effective angle of attack is computed from the specified gust and oscillation kinematics of each case

using Equation 5, and the corresponding quasi-steady lift amplitude is interpolated from a lift polar obtained from simulations of the airfoil in steady flow at angles of attack between $0°$ and $15°$. The lift phase is referenced to the phase of the effective angle of attack and a phase lag with respect to this reference is defined to be negative. The transfer-function normalization thus accounts for the differences between the aerodynamic properties of the ideal thin airfoil assumed in the theoretical framework and the simulated NACA-0012 profile, allowing us to generalize the predictions to arbitrary aerodynamic sections.



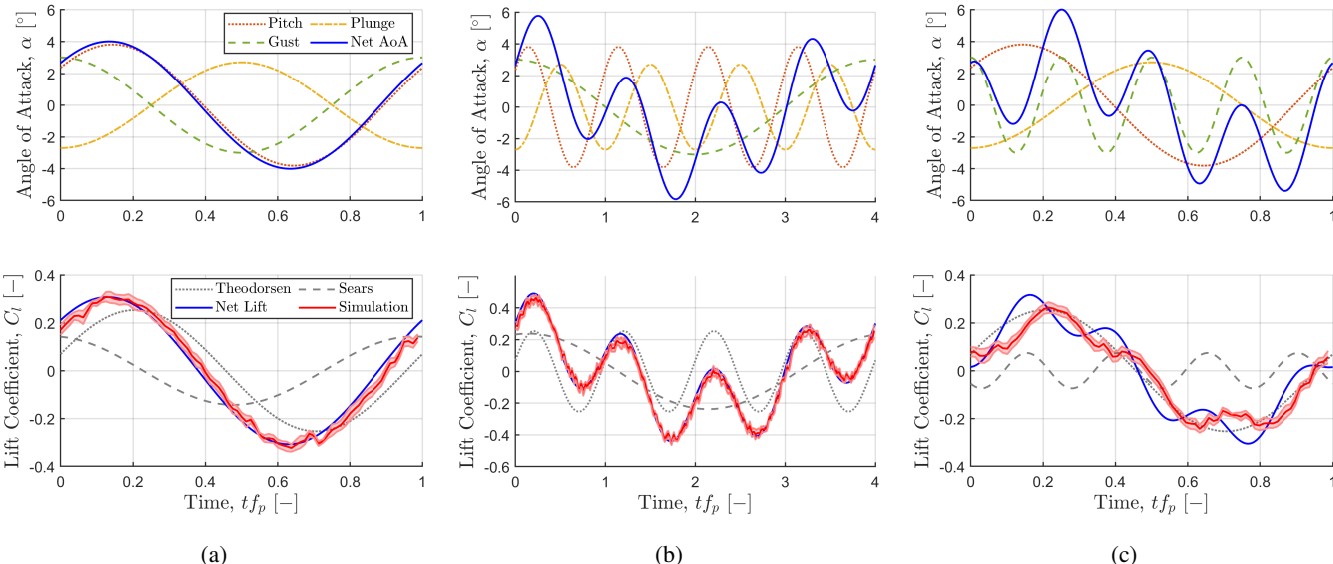

**Figure 4.** Time-varying quantities from sample cases with $f_g/f_p = 1$ (a), $f_g/f_p = 0.25$ (b), and $f_g/f_p = 4$ (c). In all cases, $\hat{\alpha}_g = \hat{\theta} = 3°$ and $\hat{h}/c = 0.03$. Top: Net angles of attack at the airfoil quarter-chord point (blue solid line) and contributions from pitch and pitch-rate (orange dotted), plunge (gold dashed-dotted), and gust (green dashed) oscillations. Bottom: Theoretical prediction of the time-varying lift coefficients (blue solid), compared with the phase-averaged simulation data (red); contributions from the Theodorsen (pitch and plunge) and Sears (gust) functions are shown as dotted and dashed grey lines.

Finally, a comparison with existing experimental results is provided to validate our approach for the specific problem of interest. Transfer-function values for the unsteady lift response of an airfoil in a sinusoidal gust are compared with the normalized Sears function (given by Equations 3 and 4) in Figure 3. The results from four simulations at two Reynolds numbers and gust-angle amplitudes are shown, in addition to experimental data from the wind-tunnel experiments of Wei et al. (2019) at similar Reynolds numbers. Good agreement is seen between the theoretical, experimental, and numerical results, suggesting that the simulation paradigm of this study is appropriate for studying the superposed gust-oscillation problem.

## 4   Results

In this section, we highlight several phase-averaged lift results to demonstrate some of the dynamical features of the problem, before considering the amplitude and phase responses of the full range of parameters treated in this study. We then present lift, transfer-function amplitude, and phase information for frequency-variation cases (FR and FRn) and amplitude-variation cases (G, PP, and PPG) to validate the theoretical framework across a wide range of unsteady conditions.



### 4.1 Angle-of-Attack and Lift Profiles

In Figure 4, we plot the computed angles of attack ($\alpha$) for three FR cases with $f_g/f_p = 1, 0.25$, and $4$, along with the gust (Sears) and airfoil-oscillation (Theodorsen) contributions to the net lift. The airfoil-oscillation reduced frequency for all of these cases is $k_p = 0.785$. The first case (Figure 4a) demonstrates a scenario in which the gust, pitching motions, and plunging motions all generate similar angle-of-attack amplitudes. In this case, the gust and plunge are nearly antiphase and thus contribute negligibly to the total angle of attack. In the second case (Figure 4b), the gust frequency is lower than the airfoil-oscillation frequency, and we observe that both the Sears and Theodorsen contributions to the net lift are needed to capture the dynamics of the problem. In both of these cases, the simulation results for the net lift agree very well with the theoretical predictions. For the third case shown in Figure 4c, the gust reduced frequency is high ($k_g = 3.142$) and the simulation data appear slightly damped, with a phase lag and decreased amplitude relative to the theoretical prediction. This could be driven by viscous effects at high reduced frequencies, which act akin to a low-pass filter. Further considerations regarding these observed discrepancies will be discussed in Section 5. Overall, these examples support the conjecture that the model can effectively capture the lift response across a range of forcing frequencies, amplitudes, and phase relations, even if some deviations appear in more extreme cases.

The time-varying lift responses from the FR cases are further explored in Figure 5, where we plot the time traces of the lift coefficient from the theoretical model against the phase-averaged simulation results. The plots are ordered by increasing frequency ratio, from $f_g/f_p = 0.5$ in Figure 5a to $5$ in Figure 5f, with Figures 4b, 4a, and 4c completing the sequence. As in the previous cases, $k_p = 0.785$. The simulation results show very good agreement with the theory at low frequency ratios, whereas a slight decrease in amplitude and increase in phase lag are observed relative to the theory as the gust reduced frequency increases above unity, again likely due to the growing influence of viscous effects not captured by the inviscid theoretical framework.

These sample cases highlight the capability of the theoretical model to satisfactorily predict the lift response from the simulation data, with the best accuracy occurring at lower reduced frequencies ($k_g \lesssim 1$). The angle-of-attack contributions from the three disturbances highlight the wide range of possible interactions between perturbation amplitudes, phases, and frequency ratios within the parameter space covered by the theoretical framework. The effective angle of attack appears to sufficiently encapsulate these rich dynamics, so that a normalization based on the quasi-steady lift amplitude may be employed as a reference for the unsteady lift data.

### 4.2 Variations of Frequency

To characterize the frequency response of the system, we investigate the effects of changing the gust frequency with respect to the airfoil-oscillation frequency on the amplitude and phase of the unsteady lift. We focus on the aforementioned **FR** and **FRn** cases, which vary the gust frequency ($f_g$) with respect to a fixed airfoil-oscillation reduced frequency of $k_p = 0.785$. The FRn cases are identical to the FR cases except that the gust waveform is shifted in phase by $180°$. In Figure 6, we plot the transfer-function amplitude and the phase, both relative to the quasi-steady lift computed from the effective angle of attack, for the theory and the simulation results. Due to complications in the definition of phase for forcing frequencies that are not integer





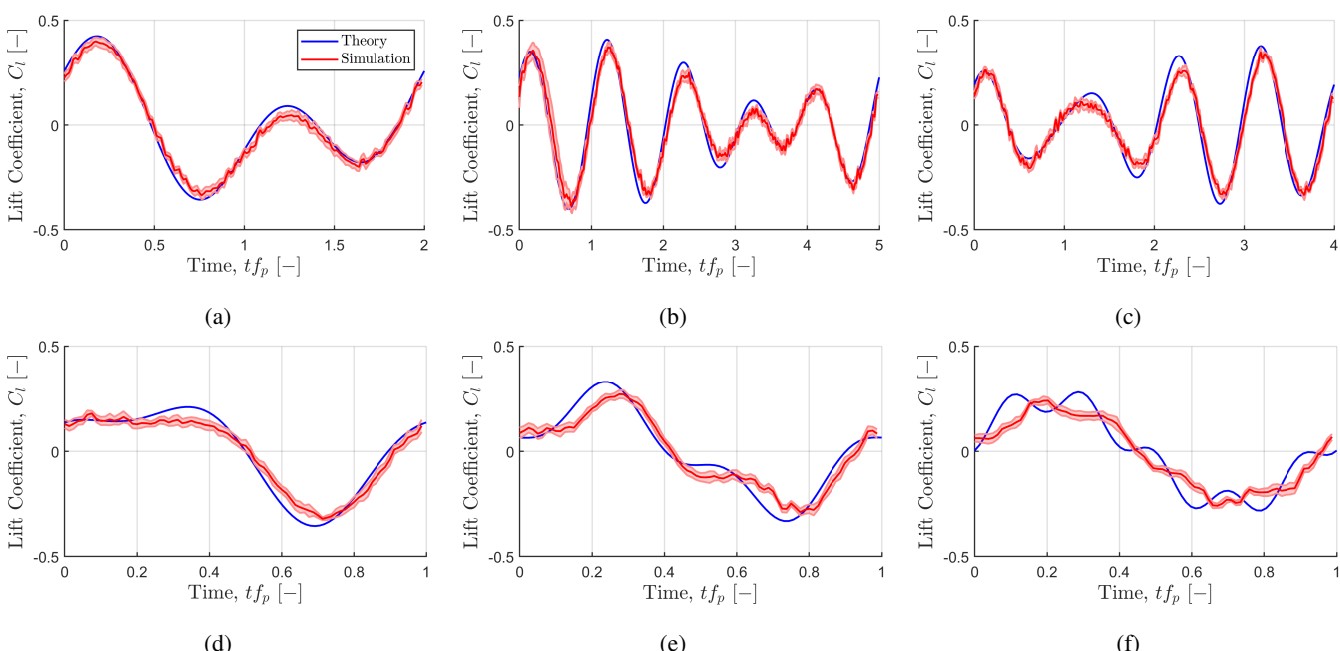

**Figure 5.** Phase-averaged lift-coefficient data for a range of frequency ratios: $f_g/f_p = 0.5$ (a), 0.8 (b), 1.25 (c), 2 (d), 3 (e), and 5 (f). $\hat{\alpha}_g = \hat{\theta} = 3°$ and $\hat{h}c = 0.03$ for all cases. Theoretical predictions are in blue and simulation results are in red. A phase lead in the theory compared to the simulations becomes more apparent at higher gust reduced frequencies (e.g. e, f).

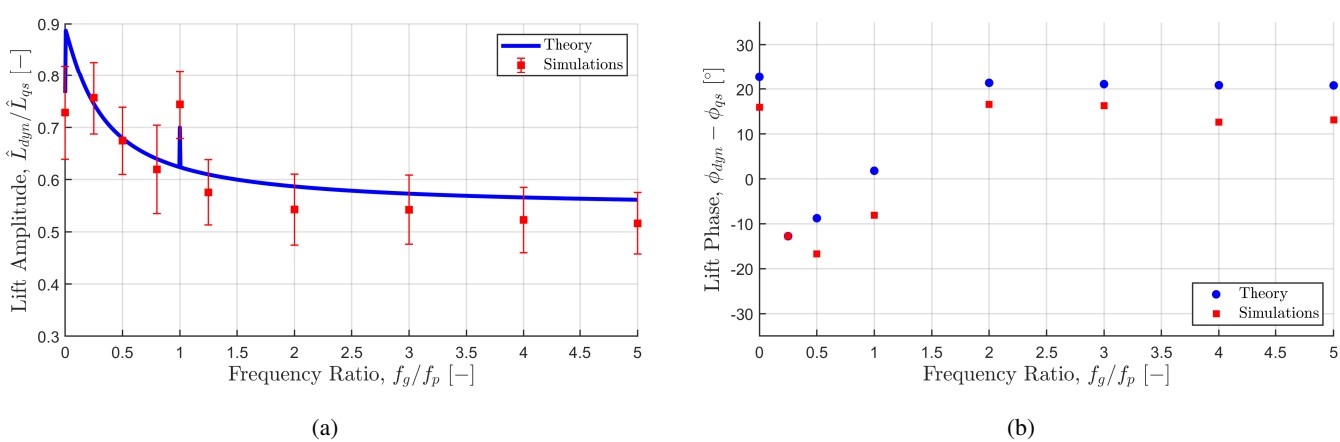

**Figure 6.** Amplitude (a) and phase (b) of time-varying lift $L_{dyn}$ for a pitching and plunging airfoil over a range of gust frequencies (FR cases), normalized by quasi-steady lift $L_{qs}$. Perturbation amplitudes held constant at $\hat{\alpha}_g = 3°$, $\hat{\theta} = 3°$, and $\hat{h}/c = 0.03$; $f_p = 1$ Hz ($k_p = 0.785$) for all cases. Theoretical predictions are in blue and simulation results are in red. Amplitude predictions are shown with a thick line to highlight discontinuous jumps at $f_g/f_p = 0$ and 1, while phase predictions are only shown at integer multiples of the gust and airfoil periods.

multiples of each other, the theoretical predictions for phase are only given at discrete points rather than as a continuous curve, and the data points at $f_g/f_p = 0.8$ and 1.25 are omitted.





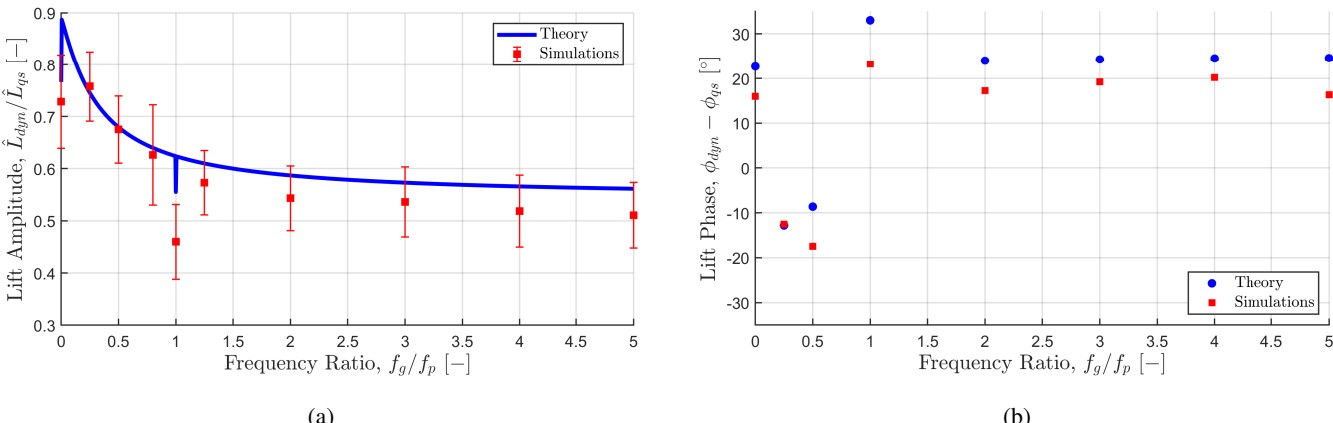

**Figure 7.** Amplitude (a) and phase (b) of the time-varying lift $L_{dyn}$ for a pitching and plunging airfoil over a range of gust frequencies (FRn cases), normalized by the quasi-steady lift $L_{qs}$. The gust phase is inverted with respect to the cases shown in Figure 6; perturbation amplitudes and frequencies are identical to these cases. Theoretical predictions are in blue and simulation results are in red.

In Figure 6a, we see that the simulation results for the amplitude closely follow the theoretical model, even for the special cases at frequency ratios of zero and unity. As discussed earlier in Section 2, the absence of a beat frequency in these special cases removes the influence of long-period variations that are present in the adjacent cases, which means that only a single phase relationship between the input forcings is sampled. This difference produces point discontinuities relative to the adjacent

frequency-ratio cases in the theoretical results, and it is thus noteworthy that these trends appear in the simulation data as well. Outside of these points, the theoretical predictions increasingly overestimate the lift amplitude obtained from the simulation data with increasing gust reduced frequency. For the phase data in Figure 6b, we see that the simulation data have a slight lag with respect to the theory, and this discrepancy remains relatively invariant across the range of frequency ratios tested.

In Figure 7, we plot the transfer-function amplitude and phase for both the theory and the simulation results for the FRn

cases. The results are very similar to Figure 6, with the sole exception being the point at $f_g/f_p = 1$, in which the inverted phase of the gust inverts the direction of the discontinuity. For the phase plot, the results are the same as for the standard FR cases. Again, the phase data at $f_g/f_p = 0.8$ and $1.25$ are omitted. Overall, the transfer-function analysis of the FR and FRn cases demonstrate that the theoretical framework is able to capture trends in the unsteady dynamics of the system in question, and therefore represents an accurate parameterization of the underlying physics.

The discontinuous behavior at unity frequency ratio and its strong sensitivity to the relative phase of the inputs make this case particularly intriguing for further study. In a fully nonlinear treatment of this problem, the interaction of forcings at matched frequencies could also lead to coupled oscillations and resonant behavior. These considerations also heighten the sensitivity of the system to high effective angles of attacks and thus flow separation. Therefore, in the following section, we consider the effects of varying the amplitudes of the disturbances at a fixed frequency ratio of 1, in order to characterize the behavior of the

system as the assumptions of the theory are challenged.

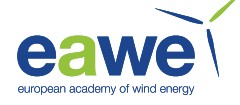


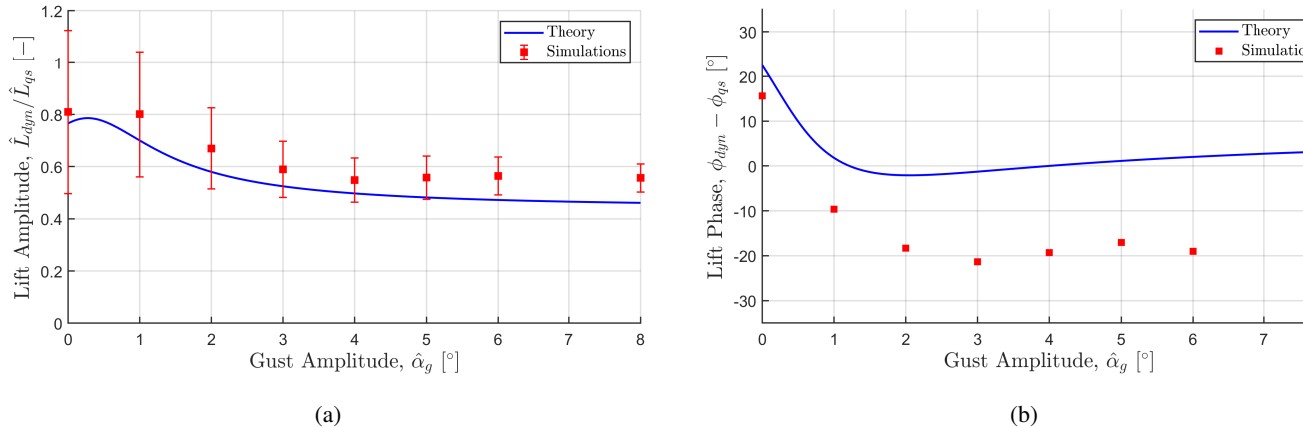

(a)  (b)

**Figure 8.** Amplitude (a) and phase (b) of the time-varying lift $L_{dyn}$ for a pitching and plunging airfoil over a range of gust amplitudes (G cases), normalized by the quasi-steady lift $L_{qs}$ and compared with theoretical predictions (blue lines). Pitch and plunge amplitudes are held constant at $\hat{\theta} = 1°$ and $\hat{h}/c = 0.01$; $f_g = f_p = 1$ Hz ($k_g = k_p = 0.785$) for all cases.

## 4.3 Variations of Amplitude

To investigate the linear response of the system, we now examine the effects of changing the gust and airfoil-oscillation amplitudes on the amplitude and phase of the unsteady lift. First, we increase the amplitude of the impinging gust, then vary the airfoil-oscillation amplitude, and finally vary all forcings simultaneously. In all of the following cases, $f_g = f_p = 1$ Hz.

In the **G** cases, the gust amplitude is varied over a fixed airfoil-oscillation frequency. This nominally matches the baseline Sears problem, but with an airfoil oscillating in small-amplitude motions. In Figure 8, we see that the theory actually underpredicts the simulation values of amplitude to some extent. At the highest values of the gust amplitude, the simulation values lie definitively above the theoretical prediction. In the corresponding phase plot, we also see that the theoretical phase shift mirrors in broad shape the simulation values, but with a discrepancy that increases with gust amplitude.

For the **PP** cases, the gust amplitude is held constant, while the pitch and plunge amplitudes are varied together. As shown in Figure 9, this means the linear theory yields a near-constant profile for airfoil-oscillation amplitudes larger than unity. The point at zero airfoil-oscillation amplitude corresponds to a pure Sears forcing. The theoretical results largely match the simulation values, with only mild underprediction of the simulation results at high airfoil-oscillation amplitudes. The theoretical predictions of the phase move toward the simulation results as the oscillation magnitude increases. This is due to the increase of

the airfoil-oscillation forces relative to the gust forces, such that the discrepancy seen in Figure 8b becomes less consequential.

In the **PPG** cases shown in Figure 10, the plunge, pitch, and gust amplitudes are all varied proportionally to each other. Since all three components are varied to have equal magnitude, the linear theory yields a constant value across the parametric range, while the simulation results vary. These numerical results for the unsteady lift amplitude are underpredicted by the linear theory. When examining the phase plot, we see that the simulation and theoretical phase difference is approximately invariant

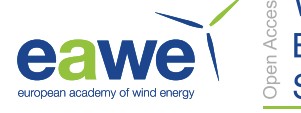

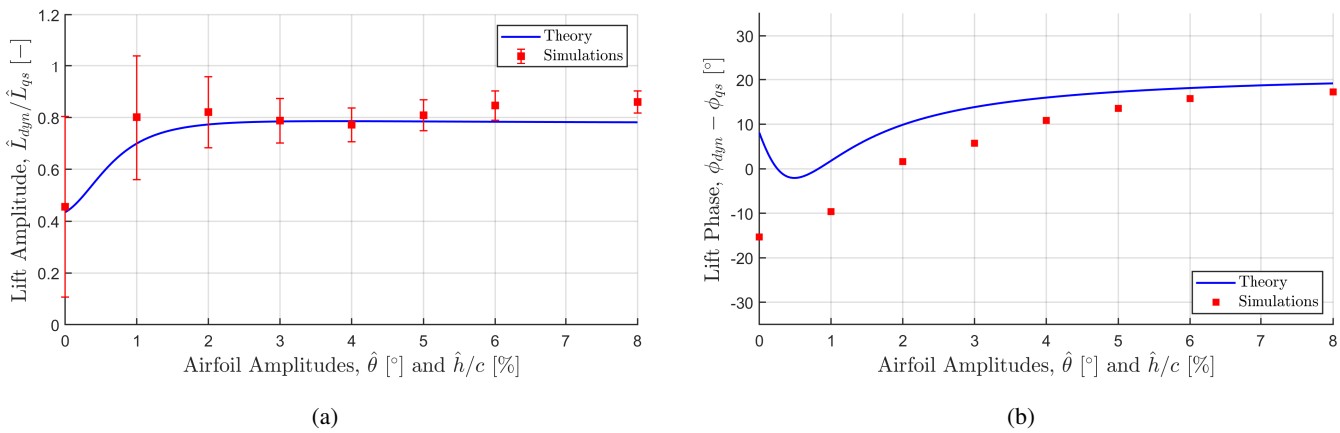

| | |
|---|---|
| (a) | (b) |

**Figure 9.** Amplitude (a) and phase (b) of the time-varying lift $L_{dyn}$ for an oscillating airfoil in a gust over a range of pitch and plunge amplitudes (PP cases), normalized by the quasi-steady lift $L_{qs}$ and compared with theoretical predictions. The gust amplitude is fixed at $\hat{\alpha}_g = 1°$, and pitch and plunge amplitudes are increased from $\hat{\theta} = 1°$ and $\hat{h}/c = 0.01$; $f_g = f_p = 1$ Hz ($k_g = k_p = 0.785$) for all cases.

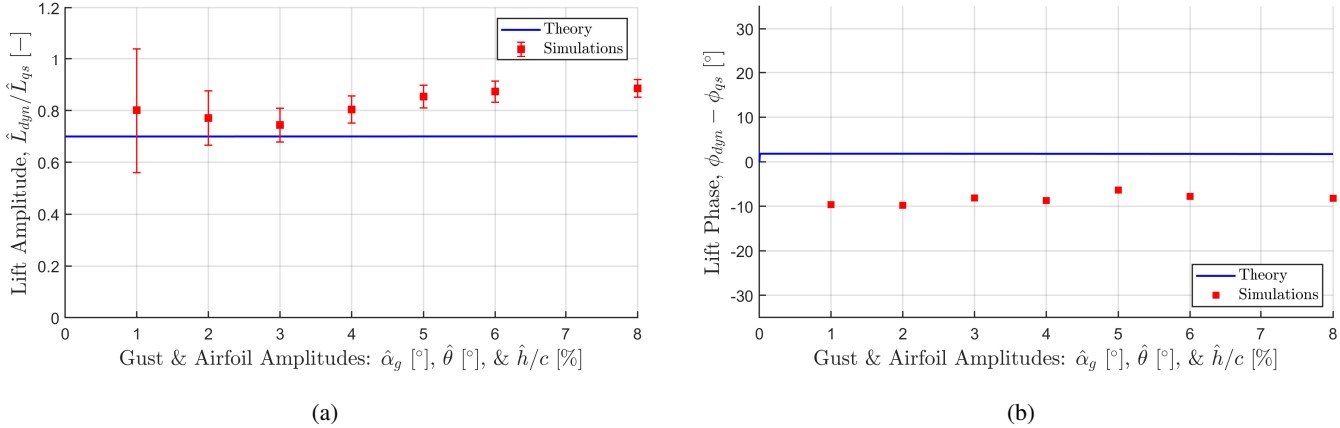

| | |
|---|---|
| (a) | (b) |

**Figure 10.** Amplitude (a) and phase (b) of the time-varying lift $L_{dyn}$ over a range of simultaneously increasing pitch, plunge, and gust amplitudes (PPG cases), normalized by the quasi-steady lift $L_{qs}$ and compared with theoretical predictions. All amplitudes are increased proportionally from $\hat{\alpha}_g = 1°$, $\hat{\theta} = 1°$, and $\hat{h}/c = 0.01$; $f_g = f_p = 1$ Hz ($k_g = k_p = 0.785$) for all cases.





across the range of amplitudes. We note here that this phase difference is closer to that observed in the G cases than the PP cases, and thus the mechanism might be most related to phase uncertainty in the gust impingement.

Part of the phase shift attributed to the differing inputs for the G, PPG, and even the FR and FRn cases may arise from uncertainty in how and when the airfoil "feels" the impinging gust. While we assume that the phase is referenced to the gust where it reaches the quarter-chord point on the airfoil, this is not an imposed condition – it is extrapolated from the advection

velocity and distance from the inlet boundary. Even if the gust reaches the quarter-chord point at the specified phase far above and below the airfoil, the nonzero thickness of the airfoil may affect the gust encounter such that the phase of the lift response does not match the idealized theoretical prediction. The airfoil pitching and plunging motions, are, however, imposed directly on the airfoil, and thus have much less uncertainty in the phase of the lift response. Therefore, it is not unexpected that cases involving a salient gust forcing (particularly the G cases) would demonstrate a phase offset relative to the theoretical results.

## 340  5  Discussion

The theoretical model proposed in this work performs remarkably well when used to predict unsteady lift, even outside the small-perturbation regime where linear theory strictly applies. As the lowest reduced frequencies ($k_g \geq 0.196$ and $k_p \geq 0.785$) sampled in this work exceed the quasi-steady limit of $k \lesssim 0.05$ proposed in the literature (Leishman, 2006), the agreement between the model and data suggests that the model sufficiently parameterizes the underlying unsteady flow physics of the

superposed gust-oscillation problem. Since reduced frequencies above $0.2$ are often considered highly unsteady in wind-energy contexts (Sebastian and Lackner, 2013), the model should yield accurate predictions for most wind-turbine applications. We also examined higher reduced frequencies than in similar investigations (e.g. Rival and Tropea, 2010; Leung et al., 2018; Taha and Rezaei, 2019), thus demonstrating the range of applicability of these potential-flow models. Still, as some deviations from the theoretical predictions in terms of the amplitude and phase of the unsteady lift signal were observed, we now consider

explanations for these discrepancies.

In Figure 11, the angle-of-attack amplitude is plotted against the error in the transfer function magnitude between the theoretical model and the simulation data, with coloring by the maximum reduced frequency (between $k_g$ and $k_p$). A few additional cases with simultaneously high forcing amplitudes and airfoil-oscillation reduced frequencies that were not presented in the previous figures are included in this plot as well. Broadly, we see that there is a correlation between angle-of-attack amplitude

and prediction error, while there does not appear to be as strong a relationship between reduced frequency and error. This implies that the dominant source of error in the modeling framework likely stems from a lack of consideration of nonlinear flow phenomena that occur at high angles of attack, such as flow separation and stall.

Neither the theoretical framework nor the simulations precisely capture stall or separation because they do not fully resolve turbulence. However, the model appears to perform well at low applied amplitudes before stall can manifest and Kitsios et al.

(2006) found that the discrepancy between 2D and 3D simulations prior to separation is minimal, roughly analogous to our regime. Furthermore, while this study is at a higher Reynolds-number condition than studied by Choi et al. (2015), that work found qualitative similarities in vortex dynamics between 2D flow simulations and 3D experiments with finite-span airfoils.





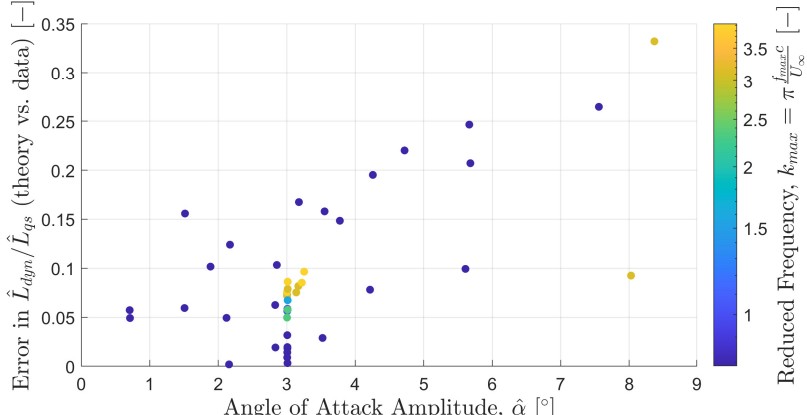

**Figure 11.** Plot of the magnitude of the error in the transfer function between the theoretical model and the simulation results versus the RMS of the effective angle of attack ($\alpha$). All cases considered in this work are shown with data points colored by the maximum of $k_p$ or $k_g$.

Negi et al. (2018) use a 2D analysis to study driving instabilities associated with transition and separation near an airfoil by invoking Squire's theorem. Accordingly, we expect our simulations to properly capture behavior that happens prior to the
growth of 2D modal instabilities that become nonlinear and beget 3D features. Even in experimental conditions, agreement with linear theory can be improved with the addition of surface-roughness or tripping elements that inhibit flow separation at relatively high angles of attack, thereby extending the predictive capabilities of the idealized theoretical framework (Wei et al., 2019). Further experiments or numerical simulations could therefore more comprehensively identify where the model starts to break down by fully resolving and manipulating separation dynamics at high forcing amplitudes and frequencies.

In addition, the simulations were all performed at a large, but finite, Reynolds number, while the theory is predicated on inviscid flow. The lower amplitude and phase differences measured in the simulation, as detailed in Section 4, follow some of the findings of Taha and Rezaei (2019). In that work, it was determined that viscous effects were a major contributor to airfoil simulations achieving lower lift amplitudes and increased phase lags relative to those predicted by theory in the pure Theodorsen problem. The viscous effects are diminished to some extent with increased Reynolds number, but since we examine
reduced frequencies at far higher values than the aforementioned work and perform eddy-resolving simulations, it is expected that similar trends would be observed in our results. The viscous-damping effect, along with the gust-phase uncertainty briefly discussed in Section 4.3, therefore can help explain the amplitude and phase discrepancies observed in the simulation data.

A complementary explanation may involve the finite thickness of the airfoil, as suggested by the theoretical analysis of Lysak et al. (2013) and the corresponding measurements of Lysak et al. (2016). Both these studies showed decreases in the
lift amplitude of an airfoil in a gust relative to the theoretical thin-airfoil prediction but did not consider viscous effects, and did not report the effects of airfoil thickness on the phase of the response. Still, the results suggest that airfoil thickness may provide a similar contribution as viscous effects to the observed discrepancies with the thin-airfoil theoretical predictions.





Finally, as we recognize that the theoretical framework by construction cannot capture nonlinear coupling or resonant effects between the two forcing modalities, it is worth noting that the amplitude data at $f_g/f_p = 1$ in the FR and FRn series showed increased deviations from the model predictions relative to the neighboring data points. These discrepancies could be due in part to the sensitivity of the amplitude to differences in the phase of the disturbances. Additionally, these deviations could also be signs of emergent nonlinear dynamics that amplify the effects of the linearly superposed signals. This possibility would also help make sense of the gradual increase in lift amplitude observed at high forcing amplitudes in the G, PP, and, particularly, the PPG cases. The highest-amplitude combination in the PPG series exhibited instantaneous angles of attack of up to $\alpha = 10.7°$, which is close to the steady-flow regime in which light stall may occur. Again, given the limitations of the 2D simulation paradigm, we cannot comment on the exact mechanisms that would lead to lift-amplitude enhancement at these high angles-of-attack. The trends in the data simply suggest that nonlinear coupling between gust and airfoil-oscillation forcings may begin to occur at unity frequency ratios and high forcing amplitudes, and that from the cases considered in this study, these conditions present the greatest likelihood for lift amplification and strong unsteady aerodynamic and aeroelastic loads.

## 6 Conclusions

This work sought to capture the combined effect of pitching and plunging motions with periodic gust perturbations on the lift of an airfoil in a single unified theoretical framework, in order to evaluate the effectiveness of analytical unsteady extensions to blade-element methods. A model derived from potential-flow theory was compared with numerical simulations of a NACA-0012 airfoil. The simulation database covered a broad range of amplitudes and frequencies for all three input forcings, and investigated reduced-frequency regimes heretofore not extensively explored. Across these cases, the model provided reasonably accurate quantitative predictions for the lift response of the airfoil. Potential mechanisms for observed differences between the theory and simulations were also discussed.

This study demonstrates the capacity of simple models of unsteady aerodynamics to capture the kinds of superpositions of gust and aeroelastic disturbances that affect wind-turbine blade sections. They provide a physics-based treatment of unsteady loads that traditional quasi-steady BEM solvers and actuator-line methods lack, with minimal added computation cost. This is of critical importance for the design and analysis of wind turbines under unsteady atmospheric flow conditions, because even at low reduced frequencies (e.g. $k \lesssim 0.2$), the difference between quasi-steady and unsteady lift amplitudes can be as much as 30% (cf. Figure 3). Therefore, even the simplest first-order parameterizations of these dynamics could yield far-reaching benefits to the operational longevity of wind turbines by better accounting for unsteady fatigue loads in the design process.

A logical next step would be to implement the unsteady modeling approach outlined here and compare the results with existing dynamic inflow models and free-vortex wake codes. This exercise would also be instructive for large-eddy simulations of wind turbines that use actuator-line models, which neglect the unsteady effects of the turbulent fluctuations they resolve on the modeled turbine aerodynamics. Furthermore, since the modeling framework explicitly accounts for the unsteady contributions to circulation from added-mass and wake effects, it should integrate well with lifting-line approaches for turbine and wake modeling and could allow for the incorporation of more complex flow interactions via additional analytical extensions. The





model transfer-function formulation may also be well-suited for the direct design of pitch and rotation-rate control schemes that could predict and mitigate unsteady loads in the linear regime in real time. The model form could also be adapted with empirical coefficients for control applications, as done by Brunton and Rowley (2013) for the pure Theodorsen problem.

The predictive nature of the theoretical framework in this study suggests that similar analytical extensions could further improve the applicability of potential-flow models in wind-energy contexts, which typically involve relatively thick airfoils at high angles of attack. Corrections for viscous effects, similar to those formulated by Taha and Rezaei (2019) to the Theodorsen function, may yield improved agreement with the data at high forcing amplitudes and frequencies. The effect of airfoil thickness would also be useful to investigate analytically along the lines of the work of Lysak et al. (2013), given the uncertainties in the gust phase noted herein. Additionally, higher-order effects, like nonzero mean angle of attack and airfoil camber, could be accounted for, as done by Goldstein and Atassi (1976) and Atassi (1984) for the Sears problem. The same theoretical approach could be extended to model the pitching moment on an oscillating airfoil in a periodic gust disturbance. These theoretical extensions would further enhance the predictive capabilities of the theoretical framework for wind-energy applications.

Due to the low perturbation amplitudes used in the simulations, the interaction of an impinging transverse gust with the dynamics of the leading-edge vortex (LEV) on the airfoil were not studied here. The LEV rolls up on an airfoil undergoing dynamic stall, and the resulting induced lift dominates the unsteady loads. The impinging gust might affect the duration for which the LEV remains bound to the airfoil, which could lead to larger unsteady lift forces due to the increased circulation of the LEV. The gust may also affect the feeding shear layer that drives the growth of circulation within the LEV, either increasing the rate of circulation accumulation or cutting off the shear layer. Future numerical and experimental studies involving high-amplitude oscillations could provide significant insights into vortex dynamics and lift control for turbine blades in dynamic stall, including the development of linear stability models for the onset of gust-induced oscillation and closed-loop control strategies for the mitigation or manipulation of vortex-induced unsteady loads on airfoils in gusty environments. Despite these limitations in the parameter space, the results of this study are sufficient to suggest that gust interactions can significantly modify the unsteady loads on an oscillating airfoil, that these unsteady dynamics should be taken seriously in the design of wind-turbine blades, and that simple analytical models can help produce longer lasting and more resilient wind turbines.

*Data availability.* The simulation software, data, and analysis scripts from this study are available upon reasonable request.

*Author contributions.* Both authors contributed equally to the conceptualization, software implementation, data analysis, and writing for this work. N.J.W. directed the theoretical component of the study, and O.B.S. led the design of the numerical setup.

*Competing interests.* The authors declare that they have no conflicts of interest.



*Acknowledgements.* The authors are grateful to Stefan Domino for extensive assistance with NaluCFD, Gianluca Iaccarino for feedback
early in the project, and preliminary discussions with Cameron Tropea and Johannes Kissing. Financial support was provided by NSF GRFP
Grant No. 1656518, Stanford Graduate Fellowships in Science and Engineering, and NSF XSEDE resources under Grant No. MCH200016.





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
