# Peer review of "Modeling unsteady loads on wind-turbine blade sections from periodic structural oscillations and impinging gusts"

_Wind Energy Science, 2023_

## Author Comment (AC1)

We thank the referee for their careful review of this manuscript and for their helpful and constructive comments. We appreciate that the referee finds this work relevant to the community, and will endeavor to incorporate their feedback in the revised manuscript. We provide the referee comments in italics and our response in standard font. Proposed manuscript changes (if substantive) are in color and describe changes made in response to each comment.

**Referee Comment:** *The classical, unsteady, linearized airfoil theories used in this study are very important in wind turbine aerodynamics, but unsteadiness is commonly ignored or represented by a quasi-steady steady process in models like blade element theory. ... There is a huge literature on unsteady behaviour of airfoils, from which the authors have drawn a comprehensive and appropriate reference list.*

We appreciate the referee's assessment and agree with the summary of the scope of this project.

**Referee Comment:** *The authors carefully state that the effects of plunging and pitching are superimposes and then on L83 describe this as a "linear model that results in a linear combination…". I think "results" is misplaced as the model is linear by construction.*

We wished to make it clear that adding two linear models maintains the linear character of the two individual components. The referee's comment suggests a more clear way to communicate this.

We have removed the first mention of linear in L83.

**Referee Comment:** *Small point: "infinite-span airfoil" is a tautology as an airfoil must have infinite span.*

We agree with the referee that it should be self-evident that our 2D setup requires an idealization, but we believe this redundant phrasing is helpful in reminding readers of this point.

**Referee Comment:** *Figure 2 shows the two-dimensional (2D) computational domain but then we are told that the unsteady vorticity field was modeled by large eddy simulation (LES) which is inherently three dimensional. How the LES is embedded in the 2D simulation is not described.*

We agree with the referee that a full large-eddy simulation (LES) is inherently three-dimensional; as such, while the nominal solver can perform LES, the two-dimensional nature of our setup and potential flow formulation of the problem

precludes this. The use of the WALE eddy viscosity model allows for a smooth transition of the 2D resolved flow to the airfoil surface, ensures that vortical motions in the near-airfoil region (where inviscid analysis is not appropriate) are not completely ignored, and stabilizes the solution at relatively large $Re_c$.

We have added clarifying language about the role of the WALE model in our simulations.

**Referee Comment:** *Small point: "to perform this transformation" on L190 is vague. I think you mean "to return to the inertial frame"?*

We have clarified L190 as suggested by the referee.

**Referee Comment:** *The discussion of the Reynolds number (Re) should be improved. My judgement is that Re > 200,000 is a good compromise as it avoids complexities like leading edge separation bubbles, that occur at lower Re while not requiring very fine grids. A vague reference to "the nonlinear effects of high Reynolds-number …" whatever they are, is not needed.*

We have expanded and revised the section of the manuscript around L190 as suggested by the referee to make the discussion clearer.

**Referee Comment:** *A brief description of the error bars in Figure 3 and the line thickness of the simulations in Figure 4 would help interpret the results. Presumably the latter represent averages over a number of cycles starting after a specified time. These details should be given. Similar remarks apply to the later figures.*

We have expanded the figure captions to better explain plotting details to the reader.

**Referee Comment:** *The effects of finite Re are mentioned briefly on L370 where the theory is said to be "inviscid". Since the classical theories contain a model for the wake and use the Kutta condition, a better adjective would be "infinite Reynolds number".*

We have added and amended the description of the flow to say it is approaching the infinite Reynolds number limit.

---

## Author Comment (AC2)

We appreciate the referee's careful review of this manuscript and helpful comments that we have used to improve the quality of the work. We provide the referee comments in italics and our response in standard font. Proposed manuscript changes (if substantive) are in color and describe changes made in response to each comment.

**Referee Comment:** *The authors have done an academic study of unsteady aerodynamics of a symmetric airfoil in inflow with transverse gust under attached flow conditions in 2D. They are considering the plunge and pitch (blade torsion) of the airfoil, and the Theodorsen effect of shed vorticity on the circulatory lift and Sears function for the unsteadiness of the lift variation from the transverse gust variation.*

We agree with the referee's summary of the scope of the work. As we will emphasize in our responses below, studying the superposition of transverse-gust and airfoil-oscillation disturbances is relevant and important to the wind energy community.

**Referee Comment:** *The authors claim that their model "could yield far-reaching benefits to the operational longevity of wind turbines by better accounting for unsteady fatigue loads in the design process." However, the authors seem to be unaware of the status of the art within aeroelastic modelling of wind turbines. The last 25 years we have had research and commercially available aeroelastic codes that include the Theodorsen effect in the unsteady aerodynamic lift models, and we have shown that these codes are able to predict the blade and turbine component fatigue loads within 5% relative error to measurements for each wind speed over the entire operational range.*

We agree with the referee that it has been shown that Theodorsen theory successfully predicts airfoil quantities far outside the strict regime it was formulated for; however, the Theodorsen function and related analytic functions are not universally utilized in the wind energy field. For example, the work of Madsen *et al.* (WES, 2020), which is used by the QBlade tool, specifically states that they do not consider Theodorsen effects. While the HAWC2 and OpenFAST modeling frameworks use semi-empirical Beddoes-Leishman inspired models to capture these effects, even research-grade large-eddy simulations (LES) of wind turbines and wind farms, such as the LESGO or Nalu-Wind codes, use actuator-disc models (ADM) and actuator-line models (ALM). Such LES employ quasi-steady aerodynamics *e.g.* a constant specified thrust coefficient (for ADM) or airfoil lift-drag polars from steady-flow measurements (for ALM).

Since turbines in LES are, by definition, exposed to inflow turbulence and time-varying disturbances, such as yaw misalignment and wind shear, the development of parsimonious unsteady models aiming to better capture the full unsteady turbine response to these factors is pertinent. A novelty of this work is examining *superposed* effects for such models, which are difficult to produce in lab experiments, and not only of

Theodorsen-type effects. To this end, we are unaware of work that examines this richer parameter space at the blade section level.

In our revised manuscript, we have tempered verbiage, better clarified our contributions, and offered commentary on limitations and advantages of our chosen problem setup.

**Referee Comment:** *The biggest uncertainty in these predictions is not the lack of transverse gust modelling (which is mainly important when the gust "wave-lengths" are of the order of the blade chord) in these codes, but the uncertainty in the inflow modelling. To capture the flapwise fatigue loads on the blades, it is very important include the deterministic components of the inflow (vertical and horizontal shear profiles, veer profiles, and yaw and upflow angles) as well as the structures of the turbulence (at least the intensity variation with height, but new methods also include turbulence reconstruction).*

We agree with the referee that inflow modeling is essential to fully characterizing wind turbine loads, but that we do not directly investigate this broader question does not diminish the relevance of our study. Indeed, atmospheric turbulence, wind shear, yaw misalignment, and wind veer represent disturbances that inform boundary conditions to our setup in the reference frame of each blade section. Even with perfect representation of inflow conditions, our study demonstrates that there is still a significant gap between the quasi-steady and unsteady lift predictions. Thus, it is important to capture unsteady aerodynamics at the blade section level in addition to inflow effects.

With respect to wind structures of the order of the blade chord that may affect turbine aerodynamics, we expect such disturbances will likely come from atmospheric boundary layer turbulence in the inflow, and will therefore appear in our modeling framework as transverse gusts (modeled by the Sears function). However, rotor-scale effects such as shear, veer, and yaw misalignment will create time-varying oscillations in the effective angle of attack at each blade section, which may be modeled by the Theodorsen function. These considerations, therefore, underscore the need for investigations of combined Sears/Theodorsen-type disturbances for wind energy applications.

In our revised manuscript conclusion section, we will more explicitly list effects that we could consider in the future to model the realistic turbine problem, including shear profiles, yaw, and stratification.

**Referee Comment:** *The authors exclude the edgewise (lead-lag) motion of the airfoil (affecting the downwash of the shed vorticity, an effect included in some aeroelastic codes). Edgewise blade vibrations due to negative aerodynamic are often driving the blade design. They are highly affected by the coupling between the edgewise airfoil motion and its pitch (blade torsion) through the lift force. An unsteady aerodynamic*

*model for wind turbines must therefore include the effect of edgewise airfoil motion on the unsteady lift.*

We appreciate the concerns of the referee and acknowledge the importance of edgewise dynamics. However, we do not intend this work to be a complete, all-inclusive model for blade design. We want to demonstrate how reduced-order potential-flow models could easily be incorporated into existing BEM / ALM simulations in order to better capture certain types of unsteady flow effects currently unaccounted for. As many base versions of these approaches assume two-dimensional sectional turbine-blade aerodynamics and do not comprehensively treat spanwise dynamics, we do the same here and leave edgewise couplings as a subject for future work.

---

## Author Comment (AC3)

We appreciate the referee's careful review of this manuscript and constructive comments that we have used to improve the quality of the work. We provide the referee comments in italics and response in standard font. Proposed manuscript changes (if substantive) are in color and describe changes made in response to each comment or groups of comments.

**Referee Comment:** *The paper proposes an unsteady aerodynamics model for combined pitching and plunging airfoil motion. The authors compare the predictions of their model, which is based in classical aerodynamics, to CFD simulations of the NACA 0012 airfoil, with some initial validation cases around the NACA 0006 airfoil. In general, the paper is written well.*

*The reviewer appreciates the idea as a potential contribution to be used in wind turbine performance codes and actuator-line methods. The originality of the idea is laudable in the sense that using concepts of classical aerodynamics to solve new problems efficiently can have notable impact.*

We generally agree with the referee's summary of the scope of the work. As we hope to emphasize in the remainder of the response, studying the superposition of transverse-gust and airfoil-oscillation disturbances, even with thin airfoils, is relevant and important to the wind energy community.

**Referee Comment:** *The NACA 0012 is irrelevant for modern utility-scale wind turbines. As there are no experimental data available (not quite certain even) for combined pitching and plunging motion, the authors should have considered a thick cambered wind turbine airfoil for comparing their model to simulations.*

**Referee Comment:** *The Discussion eludes to the fact that the model would have challenges for more relevant thick airfoils.*

While a NACA 0012 airfoil is not a cambered design, it is consistent with a potential flow setup and investigation of pre-separation behavior. We do not intend this work to be a complete, all-inclusive model for blade design; we want to demonstrate how reduced-order potential-flow models for certain types of unsteady flow effects can easily be incorporated into existing BEM/ALM simulations to capture unaccounted-for physics. This strategy mirrors the development of the Beddoes-Leishman model for dynamic stall, which was originally derived for thin airfoils. Corrections already used in the literature can make this thin airfoil work apply to a broader class of shapes; we already cite the work of Lysak *et al.* (2013, 2016) on thickness corrections in the manuscript (*cf.* lines 48, 378-379, and 422-423 in the original manuscript).

Regarding validation, we do agree with the referee that the combined setup of an oscillating airfoil in a transverse gust is difficult to match in experiment, but do respectfully disagree that "there are no experimental data available … for combined pitching and plunging motion." Pitching and plunging airfoils have been extensively explored in the literature, (e.g. Anderson et al. (1998), Rival and Tropea (2010), Baik et al. (2012)), and the problem remains of research interest. For example, a newly published experimental study by Feng and Wang (2024) examines pitching motions of a NACA 0012 airfoil in a sinusoidal transverse gust and shows good agreement between measurements and potential-flow models; however we reach Reynolds numbers and reduced frequencies inaccessible by that work that are closer to those experienced in wind energy applications.

In our revised manuscript, we have better clarified how this thin airfoil theory can be connected to thicker airfoils and add comparisons to recent work that demonstrates the relevance of our investigations.

**Referee Comment:** *The U. Glasgow database of unsteady airfoil data (among others) could have been used as a further validation case (with more appropriate airfoils) instead of a fairly recent study on the NACA 0006, which again is irrelevant for modern wind turbines.*

A novelty of this work is examining **superposed** effects for such models, which are difficult to produce in lab experiments, and not only of Theodorsen-type effects and there is scant work that examines this richer parameter space at the blade section level. However, since we do not examine the dynamic stall limits at which most unsteady airfoil data used is captured, (*e.g.*, commonly-used NREL OSU data for the NACA 4415 airfoil is at a mean angle-of-attack of 8 degrees), it would not be appropriate to directly use data from most databases as validation against our simulations. The experiments with the NACA 0006 profile were the most relevant comparisons we could find in the literature due to the airfoil's alignment with the assumptions of our potential-flow based modeling approach and the similar Reynolds numbers in the experiments. As we have argued above, models that capture the dynamic stall phenomena or thick airfoil effects can be used in conjunction with our results.

**Referee Comment:** *In general, the authors somewhat neglect decades of work being done in unsteady aerodynamics and more suitable test cases and data that would be helpful in verifying and validating their model.*

We agree with the referee that there is a long history of work in unsteady aerodynamics, and given that this study is focused on directing some of those methods of analysis to wind-energy applications, we found it impossible to reference these works exhaustively. However, we have referenced quite a few existing studies that examine related

phenomena to our specific research question (*cf.* Lines 46-54 and 66-67 in the original manuscript).

On the broader question of model development in unsteady aerodynamics, the Theodorsen function and related analytic functions are not universally utilized in the wind energy field. For example, the work of [Madsen *et al.* (WES, 2020)](), which is used by the [QBlade tool](), specifically states that they do not consider Theodorsen effects. While the [HAWC2]() and [OpenFAST]() modeling frameworks use semi-empirical Beddoes-Leishman inspired models to capture these effects, even research-grade large-eddy simulations (LES) of wind turbines and wind farms, such as the [LESGO]() or [Nalu-Wind]() codes, use actuator-disc models (ADM) and actuator-line models (ALM). Such LES employ quasi-steady aerodynamics *e.g.* a constant specified thrust coefficient (for ADM) or airfoil lift-drag polars from steady-flow measurements (for ALM).

In our revised manuscript, we have clarified the relevance of our work to the wind-energy community and offered commentary on limitations and advantages of our chosen problem setup.

**Referee Comment:** *In its present form, the reviewer cannot implement the unsteady pitching and plunging model into a BEMT code as not enough information is given. There is not even a nomenclature in the paper.*

We agree with the reviewer that this final critical application step could be better clarified in the manuscript. As we propose a modification to the standard lift coefficient calculations, this model is relatively simple to implement in a BEM code, where further corrections to account for finite thickness and camber are already present. For example, to implement in AeroDyn, which underpins OpenFAST, we would modify the static inviscid lift coefficient $C_l^{st}(\alpha)$ as it is defined in the [theory manual](). This would be similarly done in other codes and is entirely consistent; for example, HAWC2, notes that part of its unsteady aerodynamics modeling involves merging ["a thin-airfoil potential flow model ... with a dynamic stall model of the Beddoes-Leishmann type."]()

In our revised manuscript, we have better clarified exactly how one would use our results in a larger turbine simulation code.

**Referee Comment:** *It is unclear in section 3.2 which part of the rotor disk (radius, azimuth) is most affected by plunging and pitching motion.*

**Referee Comment:** *Similar in section 4.1. Where on the rotor disk of a modern wind turbine are these scenarios relevant?*

In this work, we examine the problem at the blade section level, where location on the rotor disk is an input to the model and is not explicitly necessary. However, practically speaking, we would expect pitching and plunging motions will manifest more readily further from the blade root.

We will add reference in our introduction to specific areas on the rotor disk where the pitching and plunging motions will be dominant.

---

## Author Comment (AC4)

We appreciate the referee's careful review of this manuscript and constructive comments that we have used to improve the quality of the work. We provide the referee comments in italics and response in standard font. Proposed manuscript changes (if substantive) are in color and describe changes made in response to each comment or groups of comments.

**Referee Comment:** *The manuscript presents an analytical model of the forces acting on an airfoil as it undergoes simultaneous pitching and plunging. This model is a linear combination of the Theodorsen function and the Sears function. The model results are validated against numerical simulations of a NACA 0012 airfoil.*

*Unfortunately, I do not believe that the manuscript warrants publication in Wind Energy Science, as it lacks novelty and scientific insight. The model is a linear combination of two decades-old analytical models, the Theodorsen function for a pitching airfoil and the Sears function for a plunging airfoil. In the introduction, the authors nicely list the work that has been done in this field over the past decades. It is rather trivial that such a linear combination yields a reasonable description of combined pitching and plunging in cases where non-linear effects are small.*

We generally agree with the referee's summary of the scope of the work. As we hope to emphasize in the remainder of the response, studying the superposition of transverse-gust and airfoil-oscillation disturbances, even with thin airfoils, is relevant and important to the wind energy community. It may not be surprising that the combination of the Sears and Theodorsen models yields accurate predictions for small-amplitude combined disturbances, but given the wide range of reduced frequencies and amplitudes we examine in this study, we believe it is still useful for establishing the utility of such a modeling framework and coupling it to extant models.

**Referee Comment:** *As the authors point out, the model fails when non-linearities become important, which is stated to be outside the scope of this work. However, this is exactly the regime that would have been interesting to model. In addition, the manuscript does not provide insight into the flow physics to explain the observations provided herein. Instead, the authors vaguely allude to viscous effects and flow separation, but many of the explanations are postulative and unconvincing. While it may be true that empirical models like Leishman-Beddoes provide less physical insight than analytical ones, the model presented herein breaks down for more complex flow behavior, whereas the parameter space well described by the model is also already well understood, so that little novel insight is provided.*

*Furthermore, the authors state that simplifying assumptions used for the numerical simulation limit its applicability to the small-amplitude perturbation regime. This however*

*limits the ability of the numerical simulations to serve as validation for the analytical model, since a validation should reveal when these assumptions break down. Currently, both the analytical and numerical approaches in this manuscript rely on major assumptions that do not hold true for real wind turbines, but no reliable validation is provided to evaluate these assumptions.*

While we agree with the referee that the model and numerical setup both make simplifying assumptions that may limit their immediate applicability, we disagree that such assumptions render this study irrelevant for wind-energy applications. Despite existing bodies of literature on gusts and airfoil oscillations, the parameter space of superposed disturbances of both gusts and airfoil oscillations remains very much unexplored. For example, a newly published experimental study by Feng and Wang (2024) examines pitching motions of a NACA 0012 airfoil in a sinusoidal transverse gusts and finds good agreement between measurements and potential-flow models, so it is not a settled fact that the combined effects of two linear phenomena are well predicted when flow is governed by formally nonlinear equations.

Furthermore, many experimental studies in unsteady aerodynamics, including the aforementioned work, are done with relatively low Reynolds numbers (often on the order of 10,000 for water-channel experiments) and reduced frequencies (limited by actuators). By contrast, our simulations allow us to reach higher Reynolds numbers and reduced frequencies difficult or impossible to achieve experimentally. These are closer to the regime of real wind-turbine blade sections than many other fundamental aerodynamics studies, which are often geared towards applications in biological propulsion or light aerial vehicles.

We considered it most appropriate to first focus on the small-amplitude limit where classical analytical models might be useful, so as to confirm that the theoretical framework provides accurate physical insights, before moving on to viscous and nonlinear effects. Simplifying both the theoretical and numerical approaches allows us to disambiguate these dominant underlying physics. Therefore, we believe the physical insights presented in the current work are quite relevant to the wind-energy community, even if the parameter space and assumptions involved do not exactly match those of real wind-turbine blade sections.

In our revised manuscript, we have better clarified the implementation of and relevance of our work to the wind-energy community and offered more commentary on limitations and advantages of our chosen problem setup.

**Referee Comment:** *Could you comment on the importance of the center of rotation, specifically pitching around the quarter chord vs around another point? For real wind turbines, what would be the best approximation of the rotation point?*

The Theodorsen model includes the center of rotation as a free parameter. In real wind turbines, the rotation point for a blade section will depend on the aeroelastic characteristics (e.g. bending and twisting) of the turbine blade from the blade root up to the section. As our focus in this study is to investigate aerodynamic loads and not structural deformations, we chose the quarter-chord point as a convenient reference for the airfoil-oscillation kinematics. Since the Sears problem represents the effects of a convective gust, it should be unaffected by the choice of center of rotation.

**Referee Comment:** *Could you comment to what extent it is possible or appropriate to correct your model for effects like airfoil thickness, camber, non-zero mean angle of attack and finite span? All of these are crucial in moving away from the idealized case to real application. In particular, could you comment on 3D effects and the extent to which this model holds for real wind turbines given that their blades have finite length and radially varying chord and inflow velocity vector?*

This work considers a thin NACA 0012 airfoil consistent with a potential flow setup and investigation of pre-separation behavior. Corrections already used in the literature can generalize thin-airfoil findings to a broader class of shapes: this strategy mirrors the development of the Beddoes-Leishman model for dynamic stall, which is widely used in the wind-energy community and which was originally derived for thin airfoils. We cite the work of Lysak *et al.* (2013, 2016) on thickness corrections in the manuscript (*cf.* lines 48, 378-379, and 422-423 in the original manuscript). The extension to the Sears function by Goldstein and Atassi (1976) and Atassi (1984) has been shown to account for the effects of camber and non-zero mean angle of attack (*cf.* Cordes *et al.*, 2017). Corrections for 3D effects have also been explored by Massaro and Graham (2015). Such literature demonstrates that it is possible to adapt the basic linear framework used in our study to scenarios more representative of real wind turbines and our future work may examine these corrections as a path towards a complete, all-inclusive model for blade design.

Specifically regarding 3D effects, our approach conforms to the assumptions of blade-element analysis that are commonly used for wind-turbine design and analysis. In these formulations, corrections can be added for tip losses, but generally the aerodynamics are parameterized using independent 2D blade sections, and radial variations (*e.g.* in chord length or inflow velocity) are accounted for separately for each section. Therefore, the extent to which our proposed unsteady modeling framework can account for 3D effects matches that of traditional BEM approaches.

**Referee Comment:** *Could you elaborate on what you mean when you say the Reynolds number is "low enough so that the nonlinear effects of high Reynolds-number turbulence are limited"? What are these Reynolds number effects you expect to not be*

*present, and to what extent are the simulations applicable to wind turbine blades, given that real blades operate at Re_c about an order of magnitude higher than your study?*

We agree with the referee that the wording of this sentence was unclear, and we recognize that our simulation Reynolds numbers are lower than those in utility-scale turbines. Most experimental measurements of unsteady aerodynamics, including the data used to validate our simulations, can only access $R \approx O(10^5)$. There are indeed differences in airfoil stall characteristics (both static and dynamic) that appear at higher Reynolds numbers, as shown by recent experiments done in a high-pressure wind tunnel ([Brunner *et al.*, 2021](); [Kiefer *et al.*, 2022]()). However, for disturbances below the airfoil-stall limit, increasing the Reynolds numbers approaches the inviscid-flow limit.

We will clarify our wording and justification for our examined parameter range.

**Referee Comment:** *Why do you investigate reduced frequencies up to k = 4 when you state that the most extreme cases in the real world are k = 1? And why do you not investigate k < 0.2 if that is the range typically observed in the real world? It seems that your parameter space is not directly relevant for wind turbines.*

These higher reduced frequencies can be understood as a representation of higher-order fluctuations that a real turbine blade might encounter. Reduced frequencies of $k > 1$ could be expected for blades encountering atmospheric turbulence and experiencing gust-induced oscillations at similar frequencies. These could also represent higher-order modes from a lower-frequency disturbance or oscillation. They are therefore still relevant for real-world turbine-blade dynamics.

Since unsteady dynamics tend to increase in importance with increasing reduced frequency, we expect that a model that captures unsteady forces at higher values of *k* should perform at least as well at lower values of *k*. This is shown clearly in Figure 4, where even cases with gusts and airfoil oscillations as high as $k = 0.785$ show good agreement with model predictions, allowing us to test the limits of the framework.

We will add additional explanations for our reduced-frequency range.

**Referee Comment:** *You assume sinusoidal oscillations. Can you comment on how realistic this is and how feasible it is to use this approach for more complex oscillation patterns?*

For the transfer function approach in this work, sinusoidal oscillations represent basis functions for transforming from spectral to physical space. While "real" oscillations and gusts are rarely exact sinusoids, the linear nature of the approach allows for any number of superposed forcings over a range of frequencies. Hence, any disturbance that can be represented by a finite Fourier series should be captured by the model, provided the angle-of-attack amplitudes do not incite nonlinearities.

**Referee Comment:** *In section 4, you state that the gust is felt by different parts of the airfoil at different times. However, in an incompressible flow, the gust should be felt everywhere in the flow field simultaneously. Thus the explanation is not convincing.*

We agree with the referee that one of the tenets of incompressible flow is that the pressure field is governed by a Poisson equation and responds globally to any disturbance. The gusts we consider here, however, are purely kinematic and do not involve this kind of pressure coupling. A convective gust, as considered in the Sears problem, remains incompressible, but involves velocity fluctuations that travel along at the speed of the inflow. Therefore, the airfoil does not "feel" these effects until the front edge of this disturbance encounters the airfoil, creating a local change in the angle of attack. As the gust travels along the airfoil, different parts of the airfoil will experience different flow velocities based on the local phase of the gust as it passes by each point. Hence, the airfoil does indeed experience the gust in a time-varying manner.

We will clarify our wording in referring to the gust motion boundary condition.

**Referee Comment:** *In section 5, you state that the dominant source of error of the analytical model is flow separation and stall. However, if I understand correctly, your simulations do not have separation and stall, so how can those effects explain the discrepancy between the model and the simulations? In particular, you do not exceed the static stall angle in any of your numerical simulations, so flow separation should not be the source of discrepancy.*

Our 2D numerical simulations can capture flow separation and stall, though the modeling assumptions inhibit us from making quantitative conclusions about the onset of these eventually 3D phenomena. We note that flow separation and static stall are not the same thing. Even in static contexts, flow can remain attached to the trailing edge while a local laminar separation bubble is formed such that the airfoil does not undergo deep stall despite separation occurring on its surface. These dynamics are present in our simulations; while we may not be able to make quantitative conclusions about their behavior (since we are not doing wall-resolved simulations), it is relevant to hypothesize that these local separation dynamics will impact the accuracy of the model at intermediate angles of attack.

**Referee Comment:** *In the last paragraph of the manuscript, you discuss dynamic stall. This is an entirely different topic from what is covered in this manuscript. Certainly the model you describe here, if unable to model simple non-linearities in the superposition of pitching and plunging effects, would not be able to describe the non-linear dynamics involved in dynamic stall. Therefore, the connection to this topic here does not make sense to me.*

We agree with the referee's understanding that we do not seek to model dynamic stall in this manuscript. However, dynamic stall and the earlier-discussed flow separation are

known difficulties in current models of blade section aerodynamics; our discussion aims to acknowledge that we are aware of the limitations of our approach and discuss prospective future phenomena to examine.

We will expand our flow separation and stall discussion in section 5 to clarify how we could address these phenomena.